# INTERNALIZING WORLD MODELS VIA SELF-PLAY FINETUNING FOR AGENTIC RL

## ABSTRACT

Large Language Models (LLMs) as agents often struggle in *out-of-distribution (OOD)* scenarios. Real-world environments are complex and dynamic, governed by task-specific rules and stochasticity, which makes it difficult for LLMs to ground their internal knowledge in those dynamics. Under such OOD conditions, vanilla RL training often fails to scale; we observe *Pass@k*–the probability that at least one of $k$ sampled trajectories succeeds–drops markedly across training steps, indicating brittle exploration and limited generalization. Inspired by model-based reinforcement learning, we hypothesize that equipping LLM agents with an internal world model can better align reasoning with environmental dynamics and improve decision-making. We show how to encode this world model by decomposing it into two components: state representation and transition modeling. Building on this, we introduce SPA, a simple reinforcement learning framework that cold-starts the policy via a Self-Play supervised finetuning (SFT) stage to learn the world model by interacting with the environment, then uses it to simulate future states prior to policy optimization. This simple initialization outperforms the online world-modeling baseline and greatly boosts the RL-based agent training performance. Experiments across diverse environments like Sokoban, FrozenLake, and Sudoku show that our approach significantly improves performance. For example, SPA boosts the Sokoban success rate from **25.6%** to **59.8%** and raises the FrozenLake score from **22.1%** to **70.9%** for the Qwen2.5-1.5B-Instruct model.

## 1 INTRODUCTION

Agentic reinforcement learning (RL) has become a primary framework for finetuning Large Language Model (LLM) agents (Wang et al., 2025b; Jin et al., 2025). Using ReAct-style scaffolds (Yao et al., 2023) for environmental interaction and optimizing for task rewards, this approach has been successfully applied to computer-use planning (Cao et al., 2025), tool use (Schick et al., 2023), and web search (Wu et al., 2025; Jin et al., 2025). However, the performance degrades sharply in out-of-distribution (OOD) environments that are beyond the LLM's pre-training distribution. Under such cases, the state is unfamiliar to the policy model, leading to low relative likelihood–quantified by combining the perplexity (PPL) of state descriptions and state space. As shown in Table 1, Sokoban, FrozenLake, and Sudoku exhibit state PPLs far above the Random-Guess PPL under these environments, yet ALFWorld and WebShop yield much lower state PPLs compared with the Random-Guess PPL, indicating a stronger distributional shift in the former environments.

To diagnose when LLM agents struggle to learn in OOD environments, we empirically investigate agent performance during RL finetuning and uncover a systematic divergence between two key metrics: *Pass@1*, which measures the success rate of the agent's highest-probability trajectory, and *Pass@k*, which measures the success rate across $k$ sampled trajectories, reflecting the ability to cover diverse correct solutions across multiple

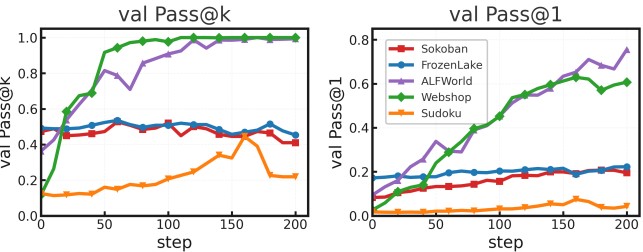

Figure 1: Validation score on five environments. Sokoban, FrozenLake and Sudoku are OOD for the models, where *Pass@k* ($k$=8) drops with training. However, in ALFWorld and WebShop, *Pass@k* increases greatly with training.

trajectories. As shown in Figure 1, in OOD environments like Sokoban, FrozenLake, and Sudoku, standard RL training causes Pass@k performance to consistently degrade, while Pass@1 increases marginally. This is different from in-domain settings (e.g., ALFWorld and Webshop), where both metrics sharply improve. This divergence reveals a fundamental limitation: current agentic RL develops a narrow, exploitative solution path (reflected in Pass@1), but fail to generalize across multiple solution paths (reflected in Pass@k). The agents can "get better at one path" but cannot build the broader world knowledge required for robust generalization in unfamiliar OOD environments. To this end, we ask an important research question: *How can we achieve effective and efficient agentic reinforcement learning in OOD environments?*

In this paper, we answer this question affirmatively. We propose `SPA` (Self Play Agent), a simple yet effective framework that enables agents to first self-play to acquire world knowledge of an OOD environment, and then exploit this knowledge to act effectively. We argue that the key to unlocking generalization of unfamiliar OOD environments is for an agent to internalize a **world model** of the environment's underlying rules, recognizing the current state and reasoning about how actions transform it to the next state over time. We refer to this structured understanding as a world model following Xing et al. (2025). The benefits of explicit world models are well-established in classical non-LLM-based RL in (Janner et al., 2019; Yu et al., 2020), where they enhance planning and sample efficiency. We hypothesize that LLM agents can benefit in an analogous way: if the model can simulate how one state changes after one action, it can reason about long action sequences in OOD environments. **World modeling enables agents to ground their reasoning in the rules of the environment rather than memorized trajectories.** However, the question of how to effectively and scalably acquire such world models for LLM agents remains open.

Our central claim is that self-play finetuning is a more effective and scalable vehicle for injecting world-model knowledge than online RL methods that inject it via reward shaping (Wang et al., 2025a). Driven by task-specific rewards, online RL is exploitative by forcing the agent to quickly narrow its policy and overfit to the first successful trajectory, improving Pass@1. Our key finding is **exploration before exploitation**: Self-play finetuning allows the agent to form a robust internal world model by exploring the environment's state space and action space, rather than overfitting reward quirks. This coverage yields a reusable scaffold that diversifies multistep reasoning and raises Pass@k.

An effective self-play finetuning, as designed in `SPA`, focuses on exploration on both state space and action space: **1) State Estimation.** To explore the state space $s_t$, the goal is to transform unfamiliar states to familiar ones through systematic exposure. When agents encounter states with high perplexity under the model's state encoder, this signals novelty and triggers agents to seek exploration via self-play and update their understanding/belief, gradually reducing their perplexity. **2) Transition Modeling.** Exploration of the action space means making the environment's transitions predictable with an accurate transition kernel $p_\theta(s_{t+1} \mid s_t, a_t)$. Effective self-play generates diverse experience triples $(s_t, a_t, s_{t+1})$ to jointly fit the state encoder and transition predictor *without* relying on dense external rewards. With the learned kernel, the reachable set under $k$ stochastic rollouts expands, directly lifting Pass@k even when Pass@1 remains low.

Putting these together, `SPA` scales improvements in RL: both Pass@1 and Pass@k grow over training. Moreover, Pass@k becomes an agentic metric, capturing whether an agent leverages a learned world model to diversify attempts and succeed within a small rollout budget, while avoiding the reward overfitting and credit-assignment brittleness common in online RL. We evaluate `SPA` on two well-studied grid worlds, *Sokoban* and *FrozenLake*, and a math-reasoning environment, *Sudoku*. These tasks demand multistep spatial reasoning and precise action control. Under a fixed compute budget of 1000 training steps, our method lifts average success rates on *Sokoban* from 25.6% to 59.8% and on *FrozenLake* from 22.1% to 70.9% as shown in Figure 2. Qualitative case studies further show that the internal world model induces more deliberate, globally consistent plans that benefit subsequent RL.

## 2 WORLD MODELING: JOINTLY OPTIMIZING STATES AND DYNAMICS

In this section, we introduce `SPA`, a simple and effective framework for improving LLM agent training. We show the framework in Figure 3. The central idea is to strengthen grounding through explicit world modeling prior to policy learning. Specifically, we decompose world modeling into two components: *state representation* and *transition dynamics*. By jointly modeling states and transition dynamics, the agent is regularized to follow structured state descriptions while interacting with the environment, thereby improving grounding. We train the world model via supervised finetuning on transition dynamics and use it to initialize policy learning. Our method does not rely on external

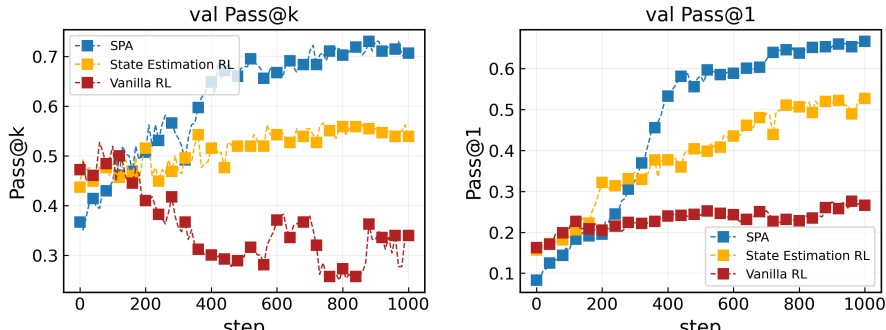

Figure 2: Validation performance on Sokoban and Frozen Lake. Red: Vanilla RL(StarPO baseline); Yellow: state-estimation RL; Blue: SPA (world-modeling SFT as a cold start, then state-estimation RL). Left: *Pass@k*; Right: *Pass@1*.

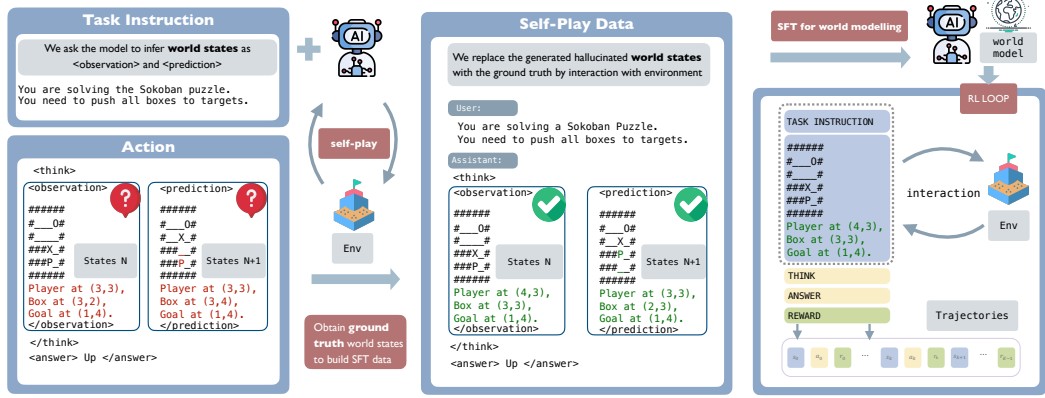

Figure 3: SPA equips an LLM agent with an inner world model. The world model is decomposed into two parts: (i) State Estimation, which converts raw observations into structured, informative natural-language descriptions of the world state; and (ii) Transition Modeling, trained in a self-play Supervised Fine-Tuning setting to predict next-state representations from current states.

knowledge or larger teacher models; instead, it learns entirely from the base model's own experience. In the following, we present the state-representation schema and the transition-modeling procedure.

## 2.1 OPTIMIZING STATE ESTIMATION

We hypothesize that a key source of difficulty in agent RL training is distribution shift: the observed states are out-of-domain with respect to the models pretraining data, making them hard to parse and reason about. In particular, LLM agents struggle to capture spatial relationships when the environment state is rendered as a symbolic text string rather than an explicit 2D layout: for instance, a 6*6 Sokoban board is represented as `######\n######\n#_###\n#_####\n#O_XP#\n######` in Wang et al. (2025b). However, when world states are expressed as raw symbols, LLMs may lose track of crucial elements, leading to unstable training. As shown in Table 1, we observe a high perplexity value of raw state representations, in contrast to the limited possible cell states. Meanwhile, environments with natural language descriptions like ALFWorld and WebShop have naturally lower PPL. To address this, we choose to provide a better and natural format (denoted as $b_t$) to abstract the

Table 1: Average PPL of state representations evaluated with Qwen2.5-Instruct-1.5B. #States denotes the number of possible values each grid cell can take(=random guess PPL). State Estimation (S Estimation) significantly reduces PPL in text games.

| Task | PPL | #States |
|---|---|---|
| Sokoban | 163.9 | 7 |
| + S Estimation | 19.6 | |
| Frozen Lake | 187.1 | 6 |
| + S Estimation | 15.4 | |
| Sudoku | 15.5 | 5 |
| + S Estimation | 7.3 | |
| ALFWorld | 6.0 | $|V|$ |
| WebShop | 11.7 | $|V|$ |

world state and concatenate it with the raw state ($s'_t$) to give a better representation. In text games, $b_t$ represents the coordinates of the positions of key game components. For example, the player, boxes, and goals in the Sokoban game. We hypothesize that using natural language descriptions can

Table 2: Results on Sokoban, FrozenLake and Sudoku for different models. (Metrics in $\times 10^{-2}$).

| Model | Sokoban | | FrozenLake | | Sudoku | |
|---|---|---|---|---|---|---|
| | Pass@1 | Pass@8 | Pass@1 | Pass@8 | Pass@1 | Pass@8 |
| Qwen2.5-0.5B-Instruct | 5.5 | 25.8 | 8.1 | 32.8 | 0.0 | 0.0 |
| +Vanilla RL | 16.9 | 35.9 | 22.8 | 30.1 | 0.0 | 0.0 |
| +State Estimation RL | 20.4 | 38.7 | 24.2 | 26.6 | 4.5 | 14.1 |
| +VAGEN (Wang et al., 2025a) | 33.3 | 44.9 | 25.8 | 40.6 | 0.1 | 0.8 |
| + SPA | 36.7 | 45.3 | 46.9 | 65.6 | 18.2 | 43.8 |
| Qwen2.5-1.5B-Instruct | 16.3 | 47.3 | 17.2 | 49.2 | 1.6 | 11.3 |
| +Vanilla RL | 25.6 | 34.0 | 22.1 | 30.7 | 0.0 | 0.0 |
| +State Estimation RL | 52.7 | 53.9 | 27.6 | 34.8 | 39.1 | 72.3 |
| +VAGEN (Wang et al., 2025a) | 44.5 | 50.0 | 37.7 | 43.0 | 0.0 | 0.0 |
| + SPA | 59.8 | 69.5 | 70.9 | 75.0 | 59.6 | 94.9 |
| Qwen2.5-3B | 12.5 | 35.9 | 6.9 | 26.9 | 0.0 | 0.0 |
| +Vanilla RL | 31.4 | 35.5 | 7.6 | 16.0 | 0.0 | 0.0 |
| +State Estimation RL | 26.2 | 27.7 | 24.7 | 35.5 | 24.1 | 26.2 |
| +VAGEN (Wang et al., 2025a) | 31.9 | 43.0 | 30.7 | 34.0 | 0.0 | 0.0 |
| + SPA | 49.7 | 58.2 | 41.3 | 46.1 | 69.9 | 89.8 |
| LLaMA3.2-1B-Instruct | 8.3 | 33.2 | 10.2 | 41.0 | 0.0 | 0.0 |
| +Vanilla RL | 21.2 | 39.8 | 10.8 | 24.6 | 0.1 | 1.2 |
| +State Estimation RL | 31.6 | 44.1 | 19.3 | 29.7 | 45.0 | 71.1 |
| +VAGEN (Wang et al., 2025a) | 47.4 | 50.8 | 24.5 | 29.3 | 0.1 | 0.8 |
| + SPA | 53.0 | 68.0 | 64.8 | 71.1 | 81.3 | 100 |
| GPT-OSS-20B | 45.8 | 84.4 | 68.8 | 100 | 61.8 | 100 |
| +State Estimation | 55.1 | 89.1 | 73.3 | 100 | 68.1 | 100 |

reduce the reasoning complexity. Finally, our optimized state representation is $s_t = \text{Concat}(s'_t, b_t)$. The table below shows an example. Text highlighted in red denotes our additions to incorporate a better state representation. In Table 1, we also observe a significant reduction in perplexity. As shown in the *State Estimation RL* row of Table 2, we see the improved representation yields higher RL performance than vanilla RL baseline (listed in the *Vanilla RL* row. Though modest, these gains indicate that better state representations benefit both Pass@k and Pass@1. In the next subsection, we will introduce how we equip the agent with transition-dynamics modeling based on state estimation.

---

**Sokoban Prompt: Symbol with Coordinates**

```
You are solving the Sokoban puzzle.
You are the player and you need to push all boxes to targets. You are provided
with a symbol grid and the zero-indexed coordinates of the player, each box, and
each target.  Coordinates range from the top-left corner (0, 0) to the
bottom-right corner (5, 5).
...
The meaning of each symbol in the state is:
#: wall, _: empty, O: target, √: box on target, X: box, P: player, S: player on
target.
Your available actions are: Up, Down, Left, Right
...
Turn 1:
State:
######
#___O#
#____#
###X_#
###P_#
######
Player (P) is at (4,3); box (X) is at (3,3); target (O) is at (1,4).
...
```

## 2.2 Injecting Transition Modeling via Self-play Fine-tuning

Based on the interpretable environment states, we inject transition dynamics into the LLM agent via a self-play finetuning process. It includes two stages: *trajectory collection via self-play* and *dynamics transition training*. During the trajectory collection stage, we use the base model to freely explore and interact with the environment under a self-play scenario to update the model's belief of the world.

**Self-Play Data Generation**   We collect the data by prompting the model to interact with the environment. This self-play interaction process with the environment is repeated over $T$ turns to collect a trajectory $\tau = (s_0, o_0, a_0, r_0, \ldots, s_{T-1}, o_{T-1}, a_{T-1}, r_{T-1}, s_T, o_T)$. At each step, the action $a_t$ is parsed and executed. The environment then produces a reward $r_t$ as feedback and transitions to a new state $s_{t+1}$, which provides the agent with a new observation $o_{t+1}$. The structure of $a_t$ is designed to incorporate explicit visual state reasoning, with a particular focus on world modeling. To reason about world states, we require the model to think before taking actions: model generation $a_t = (z_t, \hat{a}_t)$ is a sequence of text including both reasoning tokens $z_t$ and executable action $\hat{a}_t$. To incorporate world modeling into the trajectories, we prompt the LLM agent to describe the current state and predict the next state during reasoning (Wang et al., 2025a). Specifically, the reasoning variable $z_t$ between `<think>` and `</think>` is required to explicitly represent both the current state and the predicted next state: `<think><observation>`$\hat{s}_t$`</observation><prediction>`$\hat{s}_{t+1}$`</prediction></think>` `<answer>`$\hat{a}_t$`</answer>`. Then, we replace the model's beliefs about current states $\hat{s}_t$ and future states $\hat{s}_{t+1}$ with the ground-truth states $s_t$ and $s_{t+1}$ from the environment, resulting in the training trajectory: `<think><observation>`$s_t$`</observation><prediction>`$s_{t+1}$`</prediction></think>` `<answer>`$\hat{a}_t$`</answer>`. Next, we perform supervised finetuning (SFT) on these trajectories that reflect the environmental dynamics, enabling the model to learn the environment dynamics. The resulting SFT model serves as a strong initialization for downstream RL training. The prompts we use are listed under "Observation then Prediction prompts" in Appendix A.

**World Modeling.**   With the constructed trajectory $\tau_{1:T}$ above, and let $\tau_{<i}$ denote the prefix up to token $i - 1$, we optimize a simple masked token-level cross-entropy over all tokens enclosed by the special tags to capture environmental dynamics and to internalize world models in agents.

$$\mathcal{L}_{\mathrm{W}}(\theta) = -\frac{1}{\sum_{i=1}^{T} M_i} \sum_{i=1}^{T} M_i \log p_\theta(\tau_i \mid \tau_{<i}), \tag{1}$$

$$\text{where } M_i = \mathbb{1}_{[\tau_i \in (\mathrm{span}(\texttt{<think>},\texttt{</think>}) \cup \mathrm{span}(\texttt{<answer>},\texttt{</answer>}))]} \tag{2}$$

We explicitly represent both the prediction of the current state and the next state in the reasoning.

This simple world-modeling training objective is to enhance the agent's grounding capability, improving its ability to generate effective next states and answers. Consequently, the tokens corresponding to the environment's observations are masked out during the loss calculation. The resulting model serves as an effective initialization for downstream policy RL learning.

## 2.3 RL Training

After obtaining a good world model by SFT, we do Reinforcement Learning to boost the policy model. We use the Proximal Policy Optimization (PPO) (Schulman et al., 2017) algorithm for RL. Denote $\tau$ as trajectory, $u_i(\theta) = \frac{\pi_\theta(\tau_i | \tau_{<i})}{\pi_{\mathrm{old}}(\tau_i | \tau_{<i})}$ as the probability ratio between the current and old policies, and let $\tau_{<i}$ denote the prefix of token $i$. The PPO loss is defined as:

$$J^{\mathrm{PPO}}(\theta) = \frac{1}{\sum_i M_i} \sum_i M_i \cdot \min\left(u_i(\theta) A_i, \mathrm{clip}(u_i(\theta), 1 - \varepsilon, 1 + \varepsilon) A_i\right),$$

where $M_i$ is a mask that is 1 for answer tokens and 0 for contextual tokens, $A_i$ is the per-token advantage and $\varepsilon$ is a clipping hyperparameter. This objective is to optimize the policy to gain more rewards. And here we use our optimized $s_t$ (described in Section 2.1) to represent the world states to better encode the world knowledge during RL training.

## 3 EXPERIMENTS

**Experiment setting and baselines** We conduct experiments on a diverse set of models of varying sizes, including Qwen2.5-0.5B-Instruct, Qwen2.5-1.5B-Instruct, Qwen2.5-3B (Team, 2024) and LLaMA-3.2-1B (Dubey et al., 2024). Our evaluation spans multiple environments, such as Sokoban, FrozenLake, and Sudoku. Our codebase is developed based on RAGEN (Wang et al., 2025b). These text-based game environments are highly controllable; we adjust task difficulty by varying grid size to match model capacity. For Sokoban, we fix a $6 \times 6$ grid for all models. For FrozenLake, we use $4 \times 4$ for smaller models (Qwen2.5-0.5B-Instruct, Qwen2.5-1.5B-Instruct and LLaMA-3.2-1B) and $6 \times 6$ for Qwen2.5-3B to avoid ceiling effects observed with the $4 \times 4$ setting. For Sudoku, we use a $4 \times 4$ grid with six empty cells to be filled. We train with a batch size of 32 and collect eight rollouts per environment at each update step. For evaluation, we use both Pass@8 and Pass@1 metrics. We use VAGEN (Wang et al., 2025a) as our baseline. This method enables online world-model learning during RL by providing observation and prediction-based reward signals. We also evaluate a SOTA model: open-source GPT-OSS-20B (OpenAI, 2025).

**Small models could outperform strong models on text games with SPA.** Our main results are shown in Table 2. For every model we evaluated, SPA consistently outperforms State Estimation RL, which in turn outperforms vanilla RL. For instance, for Qwen2.5-1.5B-Instruct, SPA could obtain 59.8 Pass@1 score after 1000 training steps for Sokoban environment, and 70.9 Pass@1 for FrozenLake environment, which even outperforms GPT-OSS-20B model. Across other models, Qwen2.5-0.5B-Instruct and Qwen2.5-3B, SPA delivers substantial improvements over baseline methods on all our evaluated environments. This suggests our approach effectively drives exploration and consolidates environment knowledge in OOD scenarios.

**SPA outperforms the reward-based method VAGEN.** VAGEN (Wang et al., 2025a) performs world modeling by eliciting current-state observations and predicting next states, and then applies a reward to encourage accurate grounding. For a fair comparison, we reimplemented VAGEN in the same state-estimation setting as SPA. Empirically, VAGEN attains performance comparable to the Abstraction PPO baseline and yields little improvement on our testbed. In contrast, SPA delivers substantial gains. We hypothesize that this advantage stems from SPA confining world-model learning to the Supervised Fine-Tuning (SFT) stage and optimizing only the accuracy objective during RL training, thereby avoiding the multi-objective interference introduced by VAGEN. Across benchmarks, SPA consistently surpasses VAGEN, suggesting that the accuracy objective is brittle and should not be perturbed during RL training.

## 4 WHAT CONTRIBUTES TO SUCCESSFUL WORLD MODELLING?

We further disentangle the contributions of transition-model learning, grounded state representations, and multi-turn RL interaction to analyze why SPA succeeds. Our analysis highlights four critical factors: (i) *transition modeling*, which distills environment dynamics and sharpens credit assignment; (ii) *grounded state representations*, which make spatial relations explicit and enable reliable perception-action coupling; (iii) *a strong policy for world-model interaction*, which drives meaningful exploration; and (iv) *sufficient world-modeling data*, which ensures adequate coverage of the environment's dynamics. In addition, we find that multi-turn RL broadens the reasoning frontier by introducing exploration–exploitation dynamics absent in prior single-turn settings.

### 4.1 EFFECT OF TRANSITION MODELLING

**Transition-model learning is key to RL scaling. Masking the SFT loss on the current and next states results in no improvement.** To isolate the effect of world modeling supervision, we remove explicit transition (world modeling) annotations (*observation* and *prediction* spans) from the SFT corpus and replace them with a masking token *[MASKED]* and set the masking tokens' loss to 0 during SFT. Starting

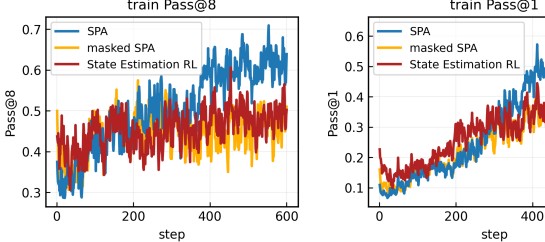

Figure 4: RL performance after world-modeling SFT under three settings: State Estimation RL baseline, No masking vs. Masking current states with next states for SPA.

from the base pretrained model, we run SFT on this masked data and use the resulting checkpoint to initialize the PPO stage. We show the results in Figure 4. In the masked setting, SFT yields no improvement for PPO. This suggests that PPOs scaling benefits derive from transition-model learning, highlighting our world-model encoding stage as pivotal to the performance.

## 4.2 Effect of Ground Truth

**Ground-truth coordinates are critical for RL training.** To test whether the abstract state format alone is sufficient, we hold the text template and training recipe fixed and replaced all ground-truth $(x, y)$ coordinates (player/box/goal) with i.i.d. random coordinates drawn from the same grid at each step. This preserves the correct format but modifies the content. In Figure 5, under this random-coordinates variant, the train-

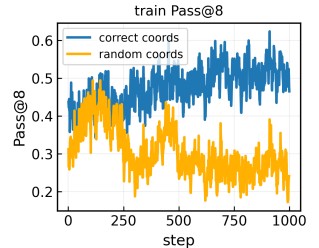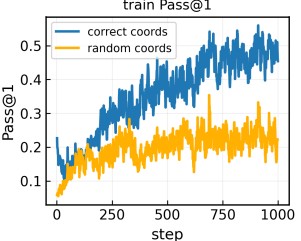

Figure 5: Pass@8 and Pass@1 comparison by using correct abstraction and incorrect abstraction on Sokoban during RL training. Left: Abstraction PPO with correct coordinates. Right: randomized coordinates (mis-specified abstraction).

ing collapses: Pass@1 and Pass@8 stagnate at low levels with fluctuation, whereas with correct coordinates, both metrics improve steadily. It shows that coordinates carry indispensable supervision for spatial grounding and credit assignment; without them, the policy cannot align with actionable spatial relations, leading to brittle exploration and poor generalization.

## 4.3 Effect of Initial Policy for Exploration

**Can the RL policy be replaced with random actions for generating world-modeling trajectories?** We realize that world-modeling trajectories need not be collected exclusively with the learned policy; they can also be obtained via generic action generators. To ablate SPA, we replace its SFT data generator with a randomaction generator: at each step, we sample uniformly from the action space Up, Down, Left, Right and substitute a fixed reasoning token sequence (*I will push the box to the target.*) As shown in Table 3, trajectories

Table 3: We replace the SFT data generator with a random-action generator and compare its performance with SPA.

| Method | Pass@1 | Pass@8 |
|---|---|---|
| Vanilla RL | 25.6 | 34.0 |
| State Estimation RL | 52.7 | 53.9 |
| SPA (1 epoch SFT) | 29.2 ↓23.5 | 52.7 ↓1.2 |
| SPA (5 epoch SFT) | 59.8 ↑7.1 | 69.5 ↑16.6 |
| SPA (1 epoch RandSFT) | 0.14 ↓52.7 | 0.39 ↓53.5 |
| SPA (5 epoch RandSFT) | 20.2 ↓32.5 | 50.0 ↓3.9 |

produced by this random policy yield substantially worse downstream RL performance, suggesting that a prior-aware policy for self-play data collection is crucial for effective world-model training to empower the subsequent RL training.

**What can we do to enable effective exploration when the policy model is weak?** We find that many failures originate from instruction noncompliance during data generation: some models fail to follow the specified output format, preventing reliable synthesis of world-modeling samples. To quantify this effect, we conduct an ablation comparing performance before and after filtering out SFT samples that deviate from the required format. Building on this, we evaluate three settings under the same RL budget on the LLaMA-3.2-1B-Instruct model in the Sokoban environment: **Original SFT**, which distills world-state transitions via supervised learning; **Filtered SFT**, where the data samples are cleaned by enforcing minimal structural validity (`<observation><prediction>` reasoning followed by an `<answer>`); and **Abs PPO**, where PPO is trained on raw grids enriched by abstraction, without transition-model SFT. As shown in Figure 8, removing misaligned data improves training stability and yields clear gains (e.g., higher Pass@k and Pass@1), indicating that strict format adherence is critical for learning accurate state and transition structure.

## 4.4 Effect of Extended SFT

To investigate the impact of spending more compute on world-modeling SFT, we vary the length of SFT from 1 to 5 epochs, and perform RL on all 5 checkpoints under the same setup. Figure 6 shows the final performance after 1000 steps of RL training. As the world model is trained longer,

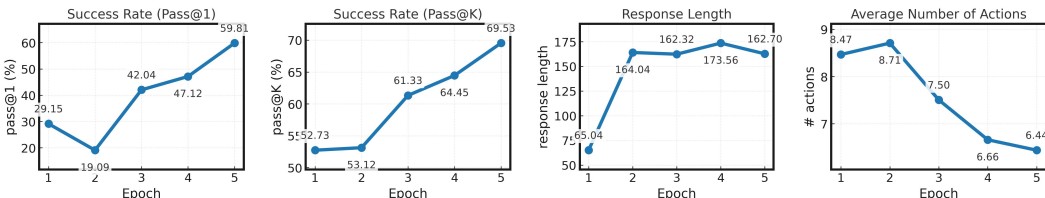

Figure 6: We evaluate RL training under initializations with varying SFT epochs and observe that increasing world-modeling SFT generally improves subsequent RL performance.

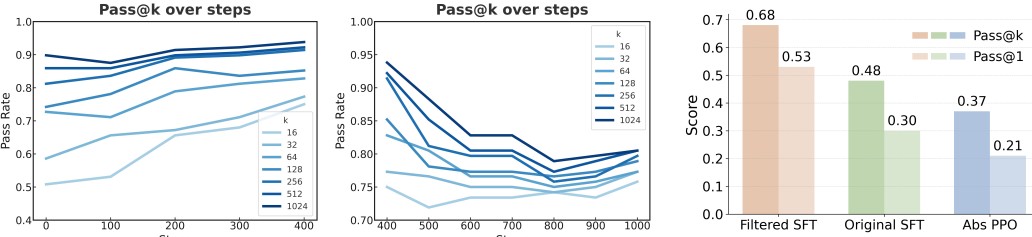

Figure 7: Pass@k performance on Sokoban during RL training. Left: steps 0–400, Pass@k consistently rises (exploration). Right: steps 400–1000, Pass@k declines as $k$ increases (exploitation). Even at k=1024, early exploration improves success rates, contrasting with RLVR findings in the math domain.

Figure 8: Results on Sokoban with LLaMA under three settings: Filtered `SPA` SFT, raw data `SPA` SFT and State Estimation PPO.

downstream RL improves consistently: Pass@1 rises from 0.29 to 0.60 and Pass@k from 0.53 to 0.70. Efficiency also improves: the average number of actions per episode decreases from 8.47 to 6.44, while action effectiveness rebounds from an early dip at Epoch 2's 0.60 to 0.77 by Epoch 5. Response length climbs from 65 tokens at Epoch 1 and then stabilizes in the 160 to 175 range for Epochs 2–5, indicating that gains are not caused by verbosity but by better spatial grounding and transition prediction. Overall, longer world-model SFT supplies a stronger dynamics prior and clearer state representations, yielding higher success and shorter trajectories under the same RL budget.

## 5 FINDINGS: EXPLORATION, EXPLOITATION, AND GENERALIZATION

In this section, we examine both the in-domain training dynamics and the out-of-domain generalization of our method, and we find that multi-turn RL reveals a distinctive exploration-exploitation trajectory and easy2hard `SPA` could work well. And the trained agents could generalize to higher task complexities within the same environment family, though it fails to transfer across games with fundamentally different dynamics.

**Environment interaction expands reasoning frontiers.** By enabling direct environment interaction, multi-turn RL broadens the reasoning frontier. Prior work (Yue et al., 2025) shows that RL with Verifiable Rewards (RLVR) improves sampling efficiency toward correct solutions but does not yield fundamentally new reasoning strategies. Moreover, RLVR exhibits a crossover in performance: RLVR-trained models outperform their base counterparts at small $k$ (e.g., $k = 1$), while base models achieve higher Pass@k at large $k$. In contrast, our multi-turn setting displays no such reversal. As shown in Figure 7, Pass@k continues to increase in the early phase of training (steps 0–400), remaining high even at $k = 1024$, and only decreases during the later phase (steps 400–1000). This trajectory reflects an exploration-then-exploitation dynamic of the policy, highlighting the necessity of environment exploration and explaining the discrepancy from math-domain RLVR results.

**World-modeling enables easy-to-hard transfer gains.** We investigate whether world modeling learned in a low-complexity setting can generalize to a more complex setting and thereby accelerate further RL training. Concretely, we train the world model with SFT on the simple FrozenLake environment (grid size $4 \times 4$) and then deploy it to boost RL on the more challenging variant ($6 \times 6$). As shown in Figure 9, the proposed easy-to-hard transfer substantially outperforms all non-world-modeling baselines, yielding faster learning and higher asymptotic returns. These results provide direct evidence that learning simple world dynamics can serve as a reusable prior that generalizes to more complex settings, boosting sample efficiency and policy quality in harder tasks.

Table 4: Results of cross-complexity and cross-game generalization experiments. The upper two rows are cross-complexity, and the lower two rows are cross-game generalization. We see that trained agents can generalize to harder environments but not to other environments.

| Evaluation setting | Pass@1 (%) | Pass@k (%) |
|---|---|---|
| Simple → Complex (Sokoban) | 0.9 | 3.1 |
| Complex (Sokoban, baseline) | 0.1 | 0.8 |
| Sokoban → FrozenLake | 15.9 | 49.2 |
| FrozenLake (baseline) | 17.2 | 49.2 |

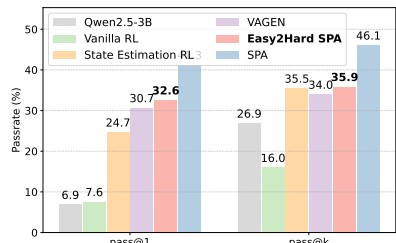

Figure 9: Results of easy-to-hard SPA on FrozenLake. Easy2Hard SPA surpasses all non-world-model baselines and VAGEN, trailing SPA trained on the target complexity.

**Generalization across games and complexity levels.** We further examine whether the trained game agents could generalize beyond its training setting. Table 4 reports two evaluation protocols. In the *cross-complexity* setting, SPA is trained on a simple Sokoban variant (6×6 grid, one box) and tested on a more complex variant (10×10 grid, two boxes). SPA outperforms a baseline trained directly on the complex task, indicating that world-model SFT facilitates scaling to higher task complexity within the same environment family. In the *cross-game* setting, SPA trained on Sokoban is evaluated on FrozenLake environment, indicating that cross-game generalization is difficult.

## 6 RELATED WORK

**OOD scenarios for LLMs.** Large language models (LLMs) (Achiam et al., 2023; Team, 2024) have advanced from static knowledge bases to versatile agents capable of planning (Cao et al., 2025), tool use (Schick et al., 2023), and multi-step reasoning (DeepSeek-AI et al., 2025). Despite these strengths, their effectiveness is largely constrained by the pre-training distribution and degrades when deployed in out-of-distribution (OOD) agents environments such as robotics (Szot et al., 2023), game playing (Shao et al., 2019), and web automation (Ning et al., 2025). These domains often involve stochastic dynamics, partial observability, and task-specific rules. Without explicit grounding in environment states, LLMs struggle to learn reliable state-conditioned policies (Xing et al., 2025), resulting in fragile or sub-optimal behavior. This grounding gap remains a fundamental limitation for deploying LLM agents in real-world, interactive scenarios.

**Agent Pipeline and Training.** To address this challenge, prior Agent work has primarily focused on prompting-based frameworks (Yao et al., 2023; Shinn et al., 2023; Topsakal & Akinci, 2023) and reinforcement learning pipelines (Song et al., 2024; Wang et al., 2025b). These approaches typically follow a short-horizon "think-then-act" paradigm, generating actions autoregressively without an internal mechanism to persist or simulate environment dynamics. In contrast, classical RL has shown the benefits of explicit world models: latent state representations and transition modeling enable stronger credit assignment, more effective planning, and higher sample efficiency (Janner et al., 2019; Yu et al., 2020). Inspired by this model-based perspective, we argue that LLM agents can similarly benefit from an internal world model. By jointly learning state estimations and transition modeling, agents can better simulate long-term consequences of actions, thereby bridging the gap between language-based reasoning and grounded policy learning. Recent work Wang et al. (2025a) introduces an online world modeling method during RL via reward shaping, which is closely related to us.

**World Modeling and Self-Play for LLMs** World models aim to learn environment dynamics to enable model-based planning (Ha & Schmidhuber, 2018; Hafner et al., 2021; Micheli et al., 2023), with recent efforts extending this idea to large language model (LLM) agents (Hao et al., 2024). While traditional self-play typically involves agents competing against themselves in multi-agent settings (Silver et al., 2017), recent LLM research has adapted this paradigm in diverse ways: SPIN (Chen et al., 2024) distinguishes self-generated samples from reference data, Self-Rewarding LMs (Yuan et al., 2024) generate intrinsic rewards, and SPAG (Pang et al., 2024) employs adversarial self-play to enhance reasoning. Our approach differs by using self-play to denote autonomous trajectory generation through environment rolloutsa usage established by MuZero (Schrittwieser et al., 2020) for single-agent Atari environments and by SelfPlayer (Nikishin et al., 2021) for single-agent exploration.

## 7 CONCLUSIONS, LIMITATIONS

LLM agents often fail to improve Pass@k in OOD environments, reflecting a reliance on memorized trajectories rather than reasoning over environment dynamics. This limitation stems from the absence of a world model that can infer world states from observations and predict future transitions. We introduce SPA, which equips agents with an internal world model by combining structured state representations with a supervised transition model and subsequent reinforcement learning via PPO. Across Sokoban, FrozenLake, and Sudoku, it outperforms prompting-based, PPO-only, and reward-grounding baselines, demonstrating that explicit world modeling middle training provides stronger dynamics priors for downstream learning. Limitations remain in stochastic settings, where training stability is fragile and instruction-following errors can compromise data quality. Future work may incorporate uncertainty-aware transitions and scale to richer modalities. SPA provides a minimal and reproducible backbone for model-based agents operating in broader interactive domains.

## ETHICS STATEMENT

All authors have read and agree to abide by the ICLR Code of Ethics.[1] This work does not involve human subjects, personal data, or sensitive demographic information, and thus does not require IRB approval. All experiments are conducted on standard, publicly available benchmark environments, which do not raise privacy, security, or fairness concerns. The methods proposed are intended for research purposes only; we do not foresee direct misuse or harmful applications beyond the standard risks associated with reinforcement learning and machine learning research in general. We declare that there are no conflicts of interest, sponsorship influences, or legal/ethical compliance issues associated with this work.

## REPRODUCIBILITY STATEMENT

We have made every effort to ensure the reproducibility of our results. The paper describes model architectures, training setups, and evaluation protocols in detail. Additional implementation details are provided in the appendix. The benchmark environments used are publicly available, and preprocessing steps are fully documented. These resources, together with the explanations in the main text and appendix, are intended to enable independent reproduction of all results reported in the paper.

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

## A    PROMPTS

**Sokoban Prompt: Base mode**

```
You are solving the Sokoban puzzle.
You are the player and you need to push all boxes to targets.
You are provided with a symbol grid and the zero-indexed coordinates of the
player, each box, and each target.
Coordinates range from the top-left corner (0, 0) to the bottom-right corner (5,
5).
When you are exactly next to a box, you can push it by moving in the same
direction.
```

```
You cannot push a box through a wall, and you cannot pull a box.
The answer should be a sequence of actions, like <answer>Right || Right ||
Up</answer>.
```

**Sokoban Prompt: Observation then prediction**

```
You are solving the Sokoban puzzle.
You are the player and you need to push all boxes to targets.
You are provided with a symbol grid and the zero-indexed coordinates of the
player, each box, and each target.
Coordinates range from the top-left corner (0, 0) to the bottom-right corner (5,
5).
When you are exactly next to a box, you can push it by moving in the same
direction.
You cannot push a box through a wall, and you cannot pull a box.
The answer should be a sequence of actions, like <answer>Right || Right ||
Up</answer>.

A sample full output is as follows:
<think>
<observation>
######
#_####
#_P###
#_X#_#
#__O_#
######
Player (P) is at (2,2); box (X) is at (3,2); target (O) is at (4,3).
</observation>
1 Down - I push box to (4,2).
2 Left - I step to (3,1).
3 Down - I stand left of box, ready to push it Right onto target.
<prediction>
######
#_####
#__###
#__#_#
#PXO_#
######
</prediction>
</think>
<answer> Down || Left || Down </answer>
```

**FrozenLake Prompt: Base mode**

```
You are solving the FrozenLake puzzle.
Forbid the whole and go to the target.
You may move to the unintended direction due to the slippery ice.
Example answer format:
<think>To forbid the hole and go to the target, I should go left then go up.
</think>
<answer>Left || Up</answer>
```

**FrozenLake Prompt: Observation then prediction**

```
You are solving the FrozenLake puzzle.
Forbid the whole and go to the target.
You may move to the unintended direction due to the slippery ice.
Example answer format:
<think>To forbid the hole and go to the target, I should go left then go up.
</think>
```

```
<answer>Left || Up</answer>

A sample full output is as follows:
<think>
<observation>
_O__
O___
G___
__P_
</observation>
Player at (3,2); holes at (0,1) and (1,0); goal at (2,0). 1 Up  move to safe ice
(2,2). 2 Left  slide to (2,1), adjacent to goal. 3 Left  reach goal (2,0);
player now on G.
<prediction>
_O__
O___
√---
----
</prediction>
</think>
<answer> Up || Left || Left </answer>"
```

**Sudoku Prompt: Base mode**

```
You are solving 4x4 Sudoku.
Fill empty cells with digits 14.
Use a 1-indexed grid (rows/cols start at 1).
A move is exactly: row,col,value (three integers).
In one turn you may output multiple moves, separated by ||.
Only propose moves that keep the row, column, and 2x2 subgrid valid. Always
output EXACTLY as:
<think>[brief reasoning]</think>
<answer>[r,c,v || r,c,v ...]</answer>
No extra text outside the two tags.
Keep the response under 50 words.
Example:
<think>Row 1 has one empty cell  place 1. Column 2 then needs 2.
</think>
<answer>1,3,1 || 3,2,2</answer>
```

**Sudoku Prompt: Observation then prediction**

```
You are solving 4x4 Sudoku.
Fill empty cells with digits 14.
Use a 1-indexed grid (rows/cols start at 1).
A move is exactly: row,col,value (three integers).
In one turn you may output multiple moves, separated by ||.
Only propose moves that keep the row, column, and 2x2 subgrid valid. Always
output EXACTLY as:
<think>[brief reasoning]</think>
<answer>[r,c,v || r,c,v ...]</answer>
No extra text outside the two tags.
Keep the response under 50 words.

An example output:
<think>
<observation>
|  . . 1 4 | 1 4 . 3 | 4 2 . . | . 1 4 2
Empty positions to be filled are at (1,1), (1,2), (2,3), (3,3), (3,4), (4,1)
</observation>
<prediction>
| 2 3 1 4 | 1 4 2 3 | 4 2 3 1 | . 1 4 2
```

Table 5: Results on Sokoban, FrozenLake, and Sudoku for different models. (Metrics in $\times 10^{-2}$).

| Model | Sokoban | | FrozenLake | | Sudoku | |
|---|---|---|---|---|---|---|
| | Pass@1 | Pass@8 | Pass@1 | Pass@8 | Pass@1 | Pass@8 |
| Qwen2.5-0.5B-Instruct | 5.5 | 25.8 | 8.1 | 32.8 | 0.0 | 0.0 |
| PPO | 16.9 | 35.9 | 22.8 | 30.1 | 0.0 | 0.0 |
| +State Estimation | 5.2 | 25.4 | 6.7 | 28.5 | 0.1 | 0.8 |
| +State Estimation RL | 20.4 | 38.7 | 24.2 | 26.6 | 4.5 | 14.1 |
| + SPA SFT | 1.8 | 11.7 | 3.1 | 15.6 | 0.1 | 0.8 |
| + SPA | 36.7 | 45.3 | 46.9 | 65.6 | 18.2 | 43.8 |
| +VAGEN | 33.3 | 44.9 | 25.8 | 40.6 | 0.1 | 0.8 |
| Qwen2.5-1.5B-Instruct | 16.3 | 47.3 | 17.2 | 49.2 | 1.6 | 11.3 |
| PPO | 25.6 | 34.0 | 22.1 | 30.7 | 0.0 | 0.0 |
| +State Estimation | 15.7 | 43.8 | 18.6 | 51.2 | 0.3 | 2.3 |
| +State Estimation RL | 52.7 | 53.9 | 27.6 | 34.8 | 39.1 | 72.3 |
| + SPA SFT | 8.3 | 36.7 | 9.8 | 43.8 | 3.5 | 24.2 |
| + SPA | 59.8 | 69.5 | 70.9 | 75.0 | 59.6 | 94.9 |
| +VAGEN | 44.5 | 50.0 | 37.7 | 43.0 | 0.0 | 0.0 |
| Qwen2.5-3B | 12.5 | 35.9 | 6.9 | 26.9 | 0.0 | 0.0 |
| PPO | 31.4 | 35.5 | 7.6 | 16.0 | 0.0 | 0.0 |
| +State Estimation | 9.9 | 39.3 | 6.9 | 26.6 | 0.1 | 0.8 |
| +State Estimation RL | 26.2 | 27.7 | 24.7 | 35.5 | 24.1 | 26.2 |
| + SPA SFT | 6.0 | 29.7 | 1.6 | 10.5 | 0.3 | 2.7 |
| + SPA | 49.7 | 58.2 | 41.3 | 46.1 | 69.9 | 89.8 |
| +VAGEN | 31.9 | 43.0 | 30.7 | 34.0 | 0.0 | 0.0 |
| LLaMA3.2-1B-Instruct | 8.3 | 33.2 | 10.2 | 41.0 | 0.0 | 0.0 |
| PPO | 21.2 | 39.8 | 10.8 | 24.6 | 0.1 | 1.2 |
| +State Estimation | 9.6 | 33.6 | 10.9 | 43.0 | 0.1 | 0.8 |
| +State Estimation RL | 31.6 | 44.1 | 19.3 | 29.7 | 45.0 | 71.1 |
| + SPA SFT | 6.3 | 27.7 | 11.4 | 38.3 | 1.3 | 9.8 |
| + SPA | 53.0 | 68.0 | 64.8 | 71.1 | 81.3 | 100 |
| +VAGEN | 47.4 | 50.8 | 24.5 | 29.3 | 0.1 | 0.8 |
| GPT-OSS-20B | 45.8 | 84.4 | 68.8 | 100 | 61.8 | 100 |
| +State Estimation | 55.1 | 89.1 | 73.3 | 100 | 68.1 | 100 |

```
Empty positions to be filled are at (4,1)
</prediction>
</think>
<answer> 1,1,2 || 1,2,3 || 2,3,2 || 3,3,3 || 3,4,1 </answer>.
```

## B  EXPERIMENTAL SETUP

Experiments run on up to $8 \times 80$G NVIDIA H100. For training, we instruction-tune on 1280 samples using AdamW with learning rate $1 \times 10^{-4}$. We apply PPO initialized from the SFT checkpoint, using a learning rate of $1 \times 10^{-6}$ for actor and $1 \times 10^{-5}$ for critic. It takes about 12 hours on single GPU to train Qwen2.5-1.5B-Instruct for 1,000 steps. For evaluation, we report decoding-only results with temperature $1.0$, top-$p$ $1.0$, and max new tokens $400$. Upon acceptance, we will release code, configuration files, and trained checkpoints.

## C  FULL RESULTS

Table 5 reports the complete results. We also include the performance before RL training (step = 0) for both settings: (i) StateEstimation PPO, evaluated with the StateEstimation prompt, and (ii) SPA immediately after the worldmodeling SFT stage. At step 0, the StateEstimation prompt yields better

Table 6: Performance of Qwen2.5-0.5B-Instruct on Alfworld.

| Method | Pass@1 | Pass@8 |
|---|---|---|
| Baseline | 30.4 | 76.5 |
| SPA | 52.1 | 93.3 |

Table 7: Performance of Qwen2.5-0.5B-Instruct on Sokoban and FrozenLake (GRPO).

| Method | Sokoban Pass@8 | FrozenLake Pass@8 | Sokoban Pass@1 | FrozenLake Pass@1 |
|---|---|---|---|---|
| Baseline RL (GRPO) | 0.0 | 0.0 | 0.0 | 0.0 |
| State Estimation RL (GRPO) | 45.9 | 34.8 | 40.5 | 31.3 |
| + SPA (GRPO) | **52.5** | **80.6** | **42.2** | **74.1** |
| Δ (Improvement) | **+6.6** ↑ | **+45.8** ↑ | **+1.7** ↑ | **+42.8** ↑ |

performance than baseline, whereas the additional worldmodeling SFT slightly lowers accuracy. However, after subsequent RL training, SPA quickly recovers and ultimately surpasses StateEstimation PPO across tasks, indicating that the structure learned during SFT provides a transferable inductive bias that RL can exploit for better generalization.

## D  USE OF LARGE LANGUAGE MODELS (LLMS)

During the preparation of this paper, we used large language models (LLMs) such as ChatGPT for assistance in polishing the writing and improving the clarity of figures. LLMs were not used for research ideation, experimental design, or the production of experimental results. All conceptual contributions, analyses, and conclusions are solely those of the authors. The authors take full responsibility for the content of this paper.

## E  SPA ON ALFWORLD

To evaluate whether SPA can generalize to more realistic settings, we also evaluate SPA on Alfworld using the Qwen2.5-0.5B-Instruct model. We choose a smaller 0.5B model as we observe that the performance quickly saturates for the 1.5B model (in Figure 1). The results are in Table 6. We observe that SPA delivers consistent gains on both the Pass@1 and Pass@k metrics. This demonstrates that our method can be effective beyond text games.

## F  SPA ON GRPO

To evaluate whether SPA can also work on other RL algorithms, we also evaluate SPA on GRPO using the Qwen2.5-1.5B-Instruct model. The results are in Table 7. We observe that SPA delivers consistent gains on both the Pass@1 and Pass@k metrics. This demonstrates that our method can be effective across the RL algorithms.

