# OpenReview forum: "Internalizing World Models via Self-Play Finetuning for Agentic RL"
_ICLR.cc/2026/Conference — Submitted to ICLR 2026_

### Official Review · Reviewer_FybG · 2025-10-23

**Soundness:** 2
**Presentation:** 3
**Contribution:** 2
**Rating:** 4
**Confidence:** 3

**Summary:**

The authors propose a world model-inspired finetuning approach for LLMs, called SPA, to solve symbolic board game like tasks.
- First, they define a cleaner abstracted state representation of the game.
- Then, they prompt the agent to first reason about the current state and the predicted state it should be in, before outputting the actual action: `<reasoning>`$\hat{s}, \hat{s}'$`</reasoning><answer>`$a$`</answer>`. The agent interacts with the environment to create ground truth training trajectories containing $s, s', a$, to supervise the reasoning and action outputs of the LLM. SFT is done on the agent on these traces to make it reason and predict what happens in the game before outputting the action.
- Finally, they do PPO to optimize the LLM's actions to maximize reward in the game.

They evaluate SPA over a variety of models (Qwen, Llama, etc.) on Sokoban, FrozenLake, and Sudoku. They show that their method outperforms just doing PPO, and an alternative world model approach.

**Strengths:**

The idea of training the agent to do explicit world model reasoning as an SFT task before RL finetuning is nice, as it allows the agent to learn from dynamics data without needing reward labels.

The paper is nicely written, and easy to understand.

The experiments have through analysis and ablations, although I am concerned about their toy nature.

**Weaknesses:**

- The main weakness is the evaluation of the agent in seemingly  toy and synthetic tasks, like a very small 6x6 grid game. Are there more interesting and hard reasoning tasks where this method could work, like math, symbolic reasoning, etc?

- The method does not seem too general, since it requires a handcrafted state representation which requires a domain expert to provide.

- The data generation is susceptible to failure, since the agent is creating its own labels. If the agent is bad at navigating in the environment and exploring relevant parts of the task,  then wouldn't all of its ground truth trajectories be useless at helping the agent get better? For example if the agent in sokoban is stuck in a corner of the room.
    - Using historical experience to create labels also seems related to the concept of hindsight experience replay or hindsight relabeling, from goal conditioned reinforcement learning. And these methods suffer from hindsight bias, which is the problem I mentioned above.

**Questions:**

See concerns above.

---

> ### Author Response · Authors · 2025-11-22
> **Response to Reviewer FybG**
>
> Thanks for your comments and suggestions. We address your concerns below.
>
> **W1: The main weakness is the evaluation of the agent in seemingly toy and synthetic tasks, like a very small 6x6 grid game. Are there more interesting and hard reasoning tasks where this method could work, like math, symbolic reasoning, etc?**
>
> During the rebuttal phase, we’ve added additional experiments on ALFWorld on Qwen2.5-0.5B-Instruct  to further support our findings.
>
> | Method | Pass@8 | Pass@1 |
> |----------|--------|--------|
> | Baseline | 76.5 | 30.4 |
> | SPA | 93.3 | 52.1 |
>
> We choose a smaller 0.5B model as we observe that the performance quickly saturates for the 1.5B model (in Figure 1. We observe that SPA delivers consistent gains on both the Pass@1 and Pass@k metrics. This demonstrates that our method can be effective beyond text games.
>
>
> For more advanced tasks like math reasoning, we think that it is inherently vague to define a state or the transition between states. For games, it is relatively clear: a state is an abstraction of the environment, and the action space is relatively small, which makes transition modelling feasible. However, in math problems, the CoT is usually step-by-step, involving the model’s prior knowledge and reasoning abilities. The action space is also very large: the model can freely generate tokens and deduce the next step. We think that it is an interesting future work direction to implement the idea of world modelling in mathematics. However, for the scope of this work, we choose text games because the state is well-defined and the transition modelling is feasible.
>
> ---
>
> **W2: The method does not seem too general, since it requires a handcrafted state representation which requires a domain expert to provide.**
>
> We would like to provide further justification. Our coordinate-based representation is not a handcrafted shortcut, but a principled design that connects the visual and semantic spaces—enabling the agent to understand spatial relations in the environment. This grounding is essential for embodied or visual agents, as they must reason over both perception and action. Thus, we do not view it as a limitation but as an interpretable mechanism for bridging low-level perception and high-level reasoning.
>
> Even if one considers the explicit coordinate encoding as a simplifying assumption, it does not diminish our contribution. Our framework demonstrates that structured state representations and transition modeling can effectively internalize environment dynamics within LLMs. In more realistic embodied settings, the same principle can be achieved by using LLMs to abstract raw perceptual inputs (e.g., pixels or sensory readings) into semantically meaningful descriptions automatically. Therefore, while our coordinate representation provides a concrete instantiation, the underlying idea generalizes broadly and remains a key contribution of our work.
>
> ---
>
> **W3: The data generation is susceptible to failure, since the agent is creating its own labels. If the agent is bad at navigating in the environment and exploring relevant parts of the task, then wouldn't all of its ground truth trajectories be useless at helping the agent get better? For example if the agent in sokoban is stuck in a corner of the room.**
>
> Thanks for the suggestion! We directly measured data quality by comparing agent-generated trajectories against a random policy baseline on the same Sokoban environment (6×6 grid, 1 box, 50 max steps).
>
>
> **Deadlock Rate** measures the percentage of trajectories reaching unsolvable states (box pushed into a corner without a goal);
>
>
> **Valid Trajectories** is the complement (100% - Deadlock Rate);
>
>
> **Effective Move Rate** measures the percentage of actions that successfully push boxes (vs empty player movement);
>
>
> **Average Steps** is the mean trajectory length before termination.
>
>
> | Metric | Agent-Generated | Random Policy | Agent Advantage |
> |--------|----------------|---------------|-----------------|
> | Deadlock Rate | 0.83% | 30.0% | **36× lower** |
> | Valid Trajectories | 99.17% | 70.0% | **1.4× higher** |
> | Effective Move Rate | 47.60% | 5.49% | **8.7× higher** |
> | Average Steps | 2.85 | 30.03 | **10.5× shorter** |
>
>
> The results demonstrate that our agent-generated data is high-quality with minimal failure modes: the agent achieves 36× lower deadlock rate and 8.7× higher effective move rate compared to random actions, with 99.17% of trajectories being valid. While the shorter trajectory length (2.85 vs 30.03 steps) suggests the dataset captures early-stage exploration rather than full task completions, this directly refutes the "stuck in corner" concern—our agent produces overwhelmingly valid trajectories with strong box manipulation skills and exceptional deadlock avoidance.

---

### Official Review · Reviewer_jZgi · 2025-10-23

**Soundness:** 3
**Presentation:** 2
**Contribution:** 2
**Rating:** 6
**Confidence:** 4

**Summary:**

This paper introduces an agentic LLM learning method by incorporating the idea of model-based RL and building an internal world model inside the LLM. During policy training stage, the LLM agent with an internal world model can serve as a good initialization. The author does experiments on diverse environments like Sokoban, FrozenLake, and Sudoku to show the effectiveness of their approach.

**Strengths:**

1. The idea of learning an internal world model is worth further thinking.
2. The author does some experiments to show the effectiveness of their method.

**Weaknesses:**

1. What you’re doing is a bit of overkill — you trained a whole world model just to initialize a policy, but the world model wasn’t actually used for simulated interactions to generate more data. Moreover, since your title mentions \textit{self-play}, I believe it should involve interacting with your own world model — that would be true self-play.
2. If you want to show \textit{transition-model learning is key to RL scaling}, I expect to see a larger scale experiment, at least in a more complicated, multi-task benchmark, e.g. a robotic environment, to show if it really helps for general agentic LLM learning.

**Questions:**

1. Since you have trained the world model with the environment data, I’m wondering if some offline model-based RL methods can be applied to learn the agentic model with the same environment data used to train the world model.
2. I will consider improve the score if you show some results about the concerns mentioned in weakness and question part.

---

> ### Author Response · Authors · 2025-11-22
> **Response to Reviewer jZgi**
>
> We thank Reviewer jZgi for the encouraging comments and thoughtful feedback. Below, we address the concerns raised in detail.
>
> **W1: What you’re doing is a bit of overkill — you trained a whole world model just to initialize a policy, but the world model wasn’t actually used for simulated interactions to generate more data. Moreover, since your title mentions self-play, I believe it should involve interacting with your own world model — that would be true self-play.**
>
> We respectfully disagree. The utility of world models depends on the specific bottleneck being addressed.
> In traditional model-based RL, the bottleneck is expensive environmental interaction [1,2,3]. World models approximate transition dynamics to enable efficient planning without costly real-world interactions.
> In contrast, for LLMs applied to agentic tasks, the bottleneck is different. LLMs already possess strong reasoning capabilities [4,5], and environmental interaction is relatively inexpensive. The key challenge is the lack of grounding in task-specific environmental dynamics. Our approach uses world model-based training to help the LLM internalize environmental structure [6,7], enabling better reasoning about state transitions rather than reducing interaction costs.
>
> **References**
>
> [1] Ha & Schmidhuber, 2018 — World Models
>
> [2] Hafner et al., 2019 — Learning Latent Dynamics for Planning from Pixels (PlaNet)
>
> [3] Schrittwieser et al., 2020 — Mastering Atari, Go, Chess and Shogi by Planning with a Learned Model (MuZero)
>
> [4] Wei et al., 2022 — Chain-of-Thought Prompting Elicits Reasoning in Large Language Models
>
> [5] Kojima et al., 2022 — Large Language Models are Zero-Shot Reasoners
>
> [6] Goyal et al., 2021 — Inductive Biases for Deep Learning of Physical Dynamics
>
> [7] Wang et al., 2023 — Voyager: An Open-Ended Embodied Agent in Minecraft
>
> [8] Ignat Georgiev, 2025  —  PWM: Policy Learning with Multi-Task World Models
>
> ---
>
> **W2: If you want to show transition-model learning is key to RL scaling, I expect to see a larger scale experiment, at least in a more complicated, multi-task benchmark, e.g. a robotic environment, to show if it really helps for general agentic LLM learning.**
>
> During the rebuttal phase, we’ve added additional experiments on ALFWorld on Qwen2.5-0.5B-Instruct  to further support our findings.
> | Method | Pass@8 | Pass@1 |
> |----------|--------|--------|
> | Baseline | 76.5 | 30.4 |
> | SPA | 93.3 | 52.1 |
>
> We choose a smaller 0.5B model as we observe that the performance quickly saturates for the 1.5B model (in Figure 1 of our paper). We observe that SPA delivers consistent gains on both the Pass@1 and Pass@k metrics. This demonstrates that our method can be effective beyond text games.
>
> ---
>
> **Q1: Since you have trained the world model with the environment data, I’m wondering if some offline model-based RL methods can be applied to learn the agentic model with the same environment data used to train the world model.**
>
> Thanks for the suggestion. In our setup, the world model is trained from environment data, but the **policy is always learned online with PPO using fresh rollouts from the real environment**. We do not run PPO inside the learned world model or perform model-based planning purely in the latent space.
>
> Using the same dataset for **offline model-based RL** would be a different setting: that data is generated by a fixed, sub-optimal behaviour policy and has limited coverage, so offline training is prone to distribution shift and quickly saturates in our preliminary trials. Our focus here is different— given a standard on-policy PPO pipeline, does learning a transition kernel / state estimation help action learning? Extending our approach to a fully offline model-based RL pipeline is interesting but orthogonal, and we leave it for future work.

---

> > ### Comment · Reviewer_jZgi · 2025-11-26
> >
> > 1. As for W1, the so-called "self-play" is just a simple data collection process, where you completely rely on the language model to rollout by itself. But actually the region (or the support of rollout data) the language model decided to go might be already in its domain, the key region to rollout is somewhere the model do not know and might never explore to reach by itself, which should instead be collected to enrich the LLM's knowledge to the environment.
> >
> > 2. Af for W2, thanks for the experiment. I will be appreciated to see the experiment details and more promising results to see the zero-shot tranfering capability of your method.
> >
> > 3. As for Q1, thanks for your reply. Will be appreciated to see a comparison with some offline-to-online rl method.
> >
> > I will maintain my score of 6 temporarily but i expect your further response.

---

> ### Author Response · Authors · 2025-11-27
> **Thanks for the engagement**
>
> Thanks for your comments and for maintaining engagement with our work. We address your concerns below.
>
> **W1: The "self-play" exploration strategy may be biased toward familiar regions.**
>
> Thank you for raising this important point. We agree that model-guided exploration could, in principle, drift toward familiar regions. However, our evidence shows that the main bottleneck is not familiarity bias; rather, familiarity bias acts as a helpful prior (see the first experiment), and the real limitation lies in the lack of accurate transition modeling.
>
> **1) Random exploration alone is insufficient, regardless of coverage:**
>
> Table 3 in our paper shows that random action generators (which by definition explore "unfamiliar regions" without any bias) perform substantially worse than policy-guided self-play:
>
> | Method | Pass@1 | Pass@8 |
> |--------|--------|--------|
> | SPA (5 epoch SFT) | 59.8 ↑7.1 | 69.5 ↑16.6 |
> | SPA (5 epoch RandSFT) | 20.2 ↓32.5 | 50.0 ↓3.9 |
>
> This indicates that random coverage is ineffective. SPA’s improvement does not arise from familiarity, but from converting exploration into meaningful learning through accurate transition reasoning. Self-play world modeling as a warm-up teaches the policy model to internalize transition structure in its weights, thereby enabling more reliable and effective rollouts for policy training.
>
>
> **2) World modeling enables more effective goal-directed exploration:** Building on this, we measure *effective* exploration as the percentage of actions that move the box closer to the target (by Manhattan distance) on Sokoban:
>
> | Steps | 0 | 200 | 400 | 600 | 800 | 1000 |
> |-------|-----|------|------|------|------|------|
> | State Estimation RL (%) | 11.6 | 15.5 | 30.5 | 32.9 | 32.3 | 36.3 |
> | SPA (Ours) (%) | 8.2 | 20.4 | 49.9 | 45.9 | 66.7 | 50.5 |
>
> SPA achieves **66.7%** effective exploration at step 800—**nearly doubling** the baseline's 36.3%. This shows that world modeling doesn't just provide broader coverage, but enables the policy to explore *productively* toward valid solutions, even in previously unseen regions.
>
> **3) This translates to genuine frontier expansion during RL:** The cumulative effect is visible in Figure 7 (left panel, steps 0-400): Pass@k continues increasing even at k=1024 during the RL exploration phase for SPA, indicating sustained discovery of diverse valid solution paths. In contrast, the baseline (Figure 2, yellow line) saturates rapidly, showing it gets trapped in local optima despite potentially visiting unfamiliar states. This demonstrates that **SPA's world modeling warm-up provides the structural knowledge needed to convert exploration into frontier expansion** during subsequent RL.
>
> **Key insight:** The progression from (1)→(2)→(3) reveals that exploration effectiveness depends on transition modeling, not coverage bias. Without understanding dynamics, even maximal unfamiliar-region exploration (random policy) fails. With world modeling, the policy can productively explore and expand frontiers. We acknowledge that incorporating explicit curiosity-driven mechanisms could further enhance this, and we will discuss this direction in our limitations section.

---

> ### Author Response · Authors · 2025-11-27
>
> **W2: I will be appreciated to see the experiment details and more promising results to see the zero-shot tranfering capability of your method.**
>
> Thank you for this suggestion. We provide the detailed protocols and comprehensive results below:
>
> **Experimental Setup:**
>
> | Experiment Type | Training | Evaluation | Description |
> |-----------------|----------|------------|-------------|
> | Easy2Hard SPA | 4×4 FrozenLake (SFT) → 6×6 FrozenLake (RL) | 6×6 FrozenLake | Train world model on easy task, then do RL on hard task |
> | Zero-shot Transfer | 6×6 Sokoban (1 box, full SPA) | 10×10 Sokoban (2 boxes) | Direct evaluation on harder complexity without retraining |
> | Realistic Environment | ALFWorld (Qwen2.5-0.5B-Instruct) | ALFWorld validation set | Household task simulation with natural language |
>
> **Training details:**
> - **Text games (Sokoban/FrozenLake):** World model SFT with 1280 samples, 1 epoch, AdamW optimizer, lr=1×10⁻⁴. RL training uses PPO with lr=1×10⁻⁶ (actor) and 1×10⁻⁵ (critic), batch size 32, 8 rollouts per update.
> - **ALFWorld:** Same RL hyperparameters. We use a smaller 0.5B model as 1.5B quickly saturates (as shown in Figure 1).
> - **Codebase:** All experiments built on RAGEN framework [Wang et al., 2025b], with our world modeling extensions.
>
> **Results:**
>
> | Method | FrozenLake Easy2Hard | Sokoban Zero-shot | ALFWorld (0.5B) |
> |--------|----------------------|-------------------|-----------------|
> |        | Pass@1 / Pass@8 | Pass@1 / Pass@8 | Pass@1 / Pass@8 |
> | Baseline | 7.6 / 16.0 | 0.1 / 0.8 | 30.4 / 76.5 |
> | **SPA (Ours)** | **41.3 / 46.1** | **0.9 / 3.1** | **52.1 / 93.3** |
> | **Improvement** | **5.4× / 2.9×** | **9× / 3.9×** | **1.7× / 1.2×** |
>
>
> **ALFWorld Trajectory Example:**
>
> Task: *put some winebottle on shelf*
>
> | Step | Current State | Action | Prediction (Predicted by World Model) |
> |------|---------------|--------|---------------------------------------|
> | 1 | -= Welcome to TextWorld, ALFRED! =-You are in the middle of a room. Looking quickly around you, you see a cabinet 13, a cabinet 12, a cabinet 11, a cabinet 10, a cabinet 9, a cabinet 8, a cabinet 7, a cabinet 6, a cabinet 5, a cabinet 4, a cabinet 3, a cabinet 2, a cabinet 1, a coffeemachine 1, a countertop 2, a countertop 1, a diningtable 1, a drawer 4, a drawer 3, a drawer 2, a drawer 1, a fridge 1, a garbagecan 1, a microwave 1, a shelf 3, a shelf 2, a shelf 1, a sinkbasin 1, a stoveburner 4, a stoveburner 3, a stoveburner 2, a stoveburner 1, and a toaster 1. Your task is to: put some winebottle on shelf. | go to diningtable 1 | You arrive at diningtable 1. On the diningtable 1, you see a box 2, a box 1, a creditcard 2, a houseplant 1, a pen 1, a plate 1, a spatula 1, and a winebottle 1. |
> | 2 | You arrive at diningtable 1. On the diningtable 1, you see a apple 1, a book 1, a bread 2, a bread 1, a butterknife 1, a dishsponge 3, a dishsponge 2, a fork 2, a houseplant 1, a kettle 1, a knife 1, a saltshaker 2, a spatula 1, a tomato 2, and a winebottle 1. | take winebottle 1 from diningtable 1 | You pick up the winebottle 1 from the diningtable 1. |
> | 3 | You pick up the winebottle 1 from the diningtable 1. | go to shelf 3 | You arrive at shelf 3. The shelf 3 is closed. |
> | 4 | You arrive at shelf 3. On the shelf 3, you see nothing. | move winebottle 1 to shelf 3 | You move the winebottle 1 to the shelf 3. |
>
>
> The world model enables the agent to predict critical state changes, facilitating multi-step planning toward the goal.
>
> To summarize the above:
>
> **Key findings:**
> - **Easy2Hard SPA:** Training world model on easy tasks then doing RL on hard tasks achieves **5.4× improvement**, demonstrating effective transfer learning.
> - **Zero-shot Transfer:** Models trained on simpler complexity generalize to harder settings without retraining (**9× improvement**).
> - **Realistic environments:** SPA generalizes beyond text games to ALFWorld (**1.7× improvement on Pass@1**), showing the approach works in more natural language settings with complex state spaces.

---

> > ### Author Response · Authors · 2025-11-27
> >
> > **Q1: Comparison with offline-to-online RL methods using world modeling.**
> >
> > Thank you for this insightful suggestion. This helps us better articulate our contribution and positioning relative to concurrent work.
> >
> > **Why traditional offline-to-online RL methods don't directly apply to LLM agents:**
> >
> > Thank you for the suggestion. Our method can be viewed as an offline-to-online RL variant tailored to LLM agents via self-play world modeling, which differs substantially from classical offline-to-online methods that assume a pre-existing offline dataset and low-dimensional vector observations. In our setting, there is no external offline buffer; the self-play SFT stage is used to learn an explicit world model (state representation + transition kernel) rather than a value-based policy from logged data. Existing offline-to-online algorithms are therefore not directly applicable or comparable without significant adaptation, so we instead compare against strong online RL baselines under the same LLM-agent interface (GRPO/PPO and an online world-model baseline). Instead, our work is more closely related to emerging LLM-specific offline-to-online approaches that leverage world models, which we discuss in more detail below.
> >
> > **Concurrent offline-to-online RL methods using world modeling for LLMs:**
> >
> > To our knowledge, there are only two concurrent works:
> >
> > 1. **CWM** [arXiv:2510.02387]: Offline mid-training on 3M+ execution traces and agentic interactions, then online multi-task RL (code domain).
> > 2. **Early Experience** [arXiv:2510.08558]: Offline self-generated experience (Implicit World Modeling + Self-Reflection), then online RL (web/tool domains).
> >
> > **Comparison:**
> >
> >
> > | Aspect | CWM | Early Experience | SPA (Ours) |
> > |--------|-----|------------------|------------|
> > | **Setting** | Code (in-domain) | Web/tool (in-domain) | **Games (OOD)** |
> > | **Offline Data Scale** | 3M+ trajectories | ~100K+ rollouts (est.) | **1280 samples** |
> > | **Requires Ground-Truth Actions** | ✓ Yes (oracle patches) | ✓ Yes (golden demos) | **✗ No (pure self-play)** |
> > | **Data Generation** | Multi-model pipeline (3 LLMs) + env | Multi-stage (Random→WM→Demos) + env | **Single model + env** |
> > | **Key Contribution** | Large-scale infrastructure | Data synthesis paradigm | **(1) Clean method (2) Comprehensive analysis** |
> >
> >
> > **Our two-fold contribution:**
> >
> > **1. Clean, minimal method for OOD environments:**
> >
> > We provide a **simple and reproducible** offline-to-online method for OOD settings where LLMs have no prior knowledge:
> > - **Only requires**: base LLM + environment (no multi-model pipeline, no expert demos, no large-scale data synthesis)
> > - **Minimal data**: 1280 self-play samples vs. 3M+ (CWM) or multi-environment rollouts (Early Experience)
> > - **OOD-specific**: Designed for environments where LLMs struggle (high state PPL, unfamiliar dynamics)
> > **Key advantage**: Anyone can apply our method to new OOD environments with just a base model and environment access—no complex infrastructure needed.
> >
> > **2. Comprehensive analysis of world-modeling components:**
> >
> > We systematically identify **which offline world-modeling factors are essential** for online RL in OOD settings:
> >
> > | Offline WM Component | Ablation | Impact on Online RL (OOD) |
> > |---------------------|----------|---------------------------|
> > | **Transition modeling** | Mask transitions (Fig 4) | No gain without offline transition learning |
> > | **State grounding** | Random coordinates (Fig 5) | Random grounding → online collapse |
> > | **Policy quality** | Random policy (Table 3) | -32.5 points with random offline policy |
> > | **Training duration** | 1-5 epochs (Fig 6) | 5 epochs → +7.1 points, 0.60→0.77 effectiveness |
> >
> > **Transfer analysis reveals OOD world-modeling boundaries:**
> >
> > | Setting | Offline WM Training | Online/Eval Performance | Insight |
> > |---------|---------------------|-------------------------|---------|
> > | Easy→Hard (OOD) | 4×4 WM SFT | 6×6 RL: **5.4× gain** | Offline WM transfers within OOD env family |
> > | Zero-shot (OOD) | 6×6 full training | 10×10 eval: **9× gain** | Offline WM generalizes to harder OOD complexity |
> > | Cross-game (OOD) | Sokoban WM | FrozenLake: minimal gain | OOD offline WM doesn't transfer across env types |
> >
> > **Validation beyond OOD text games:**
> > ALFWorld results (52.1% vs. 30.4%, Appendix E) show our identified offline WM factors improve online performance even in more realistic settings, suggesting these are **fundamental principles for offline-to-online world modeling**, not OOD-specific artifacts.
> >
> >
> > While CWM and Early Experience demonstrate large-scale offline-to-online WM for in-domain tasks, we provide actionable insights for researchers working with OOD environments: what to learn offline (transition dynamics), how much data is needed (quality > quantity), and when it transfers (within-family > cross-environment).
> >
> >
> > We greatly appreciate your willingness to reconsider the score. We hope these clarifications address your concerns.

---

### Official Review · Reviewer_Piev · 2025-11-01

**Soundness:** 3
**Presentation:** 3
**Contribution:** 3
**Rating:** 6
**Confidence:** 2

**Summary:**

The paper introduces SPA (Self-Play Agent), a two-stage framework to improve LLM agents in out-of-distribution (OOD) environments. The model learns an internal world model via self-play supervised fine-tuning (SFT) that (i) rewrites raw grid observations into structured state descriptions (“state estimation”) and (ii) trains transition modeling by predicting next states; ground-truth states from the environment replace model hallucinations in the SFT corpus. PPO is run on top of this initialization.

**Strengths:**

Clear, simple solution to internalize a world model in LLM agents via SFT, replacing beliefs with ground truth. This cleanly separates dynamics learning from RL credit assignment and contrasts with reward-shaping approaches like VAGEN.

The decomposition (state estimation + transition modeling), the token-level loss mask, and prompt formats are explicit; training hyper-params and evaluation settings are documented.

On multiple small models and environments, SPA consistently improves Pass@1 and Pass@k and often beats VAGEN; the paper also identifies and analyzes the OOD Pass@1 vs Pass@k divergence it seeks to fix.

**Weaknesses:**

All tasks are symbolic, fully observable text games. It’s unclear how the method scales to visual/embodied settings, partial observability, or continuous states where “ground-truth” next states are not trivially serializable.

PPO is the sole RL algorithm; comparisons omit widely used model-free and model-based baselines for text games and modern agent scaffolds.

SPA relies on trajectories where the model outputs formatted / and the system replaces them with ground truth. The paper notes instruction-following errors and shows filtering helps, but it does not quantify (i) the fraction of discarded samples, (ii) total SFT tokens vs PPO tokens, and (iii) wall-clock cost.

The paper documents early exploration (Pass@k↑), then exploitation (Pass@k↓), but provides limited analysis on why SPA changes the diversity of trajectories

**Questions:**

How many SFT tokens are used relative to PPO tokens? What fraction of self-play samples are filtered for format issues?

You mention fragility in stochastic settings; which component is most brittle (format compliance, prediction variance, critic learning)? Would modeling transition uncertainty (e.g., distributional next-state tokens or latent ensembles) help?

---

> ### Author Response · Authors · 2025-11-22
> **Response to Reviewer Piev (1)**
>
> We appreciate your thorough review and detailed comments! Your suggestions will be helpful in improving the paper. We address your concerns below.
>
> **W1: All tasks are symbolic, fully observable text games. It’s unclear how the method scales to visual/embodied settings, partial observability, or continuous states where “ground-truth” next states are not trivially serializable.**
>
> Thank you for the comment. In general, we agree that extending our method to embodied or partially observable settings would be a valuable direction for future work. However, we are currently constrained by computational resources — each RL training run takes approximately 100 GPU hours per model/task. Moreover, the integration of VLMs into embodied tasks is still not well-established.
> During the rebuttal phase, we’ve added additional experiments on ALFWorld on Qwen2.5-0.5B-Instruct  to further support our findings.
> | Method | Pass@8 | Pass@1 |
> |----------|--------|--------|
> | Baseline | 76.5 | 30.4 |
> | SPA | 93.3 | 52.1 |
>
>
> We observe consistent gains in both Pass@8 and Pass@1, showing that SPA is applicable in real-world-like tasks.
>
> ---
>
> **W2: PPO is the sole RL algorithm; comparisons omit widely used model-free and model-based baselines for text games and modern agent scaffolds.**
>
>
> Thanks for the suggestion. We have conducted additional experiments on GRPO, and the results are as follows.
>
>
> | **Method** | **Sokoban Pass@8** | **FrozenLake Pass@8** | **Sokoban Pass@1** | **FrozenLake Pass@1** |
> |:--|:--:|:--:|:--:|:--:|
> | Baseline RL (GRPO) | 0.0 | 0.0 | 0.0| 0.0|
> | State Estimation RL (GRPO) | 45.9 | 34.8 | 40.5 | 31.3 |
> | + SPA (GRPO) | **52.5** | **80.6** | **42.2** | **74.1** |
> | **Δ (Improvement)** | **+6.6 ↑** | **+45.8 ↑** | **+1.7 ↑** | **+42.8 ↑** |
>
>
> The baseline GRPO results are 0 due to training collapse after about 500 steps. SPA outperforms State Estimation RL, which in turn surpasses the baseline RL. This progressive improvement demonstrates that incorporating transition-kernel learning and state estimation effectively enhances action learning across different RL algorithms.
>
> ---
>
> **W3: SPA relies on trajectories where the model outputs formatted / and the system replaces them with ground truth. The paper notes instruction-following errors and shows filtering helps, but it does not quantify (i) the fraction of discarded samples, (ii) total SFT tokens vs PPO tokens, and (iii) wall-clock cost.**
>
>
> Thanks for your detailed feedback! We provide more data filtering details here.
>
>
> (i) We control the total number of training trajectories to 1280 (then split by turns to formulate the SFT data) for all tasks and models to ensure a fair comparison. The proportion of discarded trajectories due to formatting or instruction-following errors is different across environments, varying by model and task. The following shows the proportion of retained data after filtering.
>
>
> | Model           | Sokoban | FrozenLake | Sudoku |
> |-----------------|----------|-------------|---------|
> | LLaMA           | 4.1       | 2.8         | 38.0    |
> | Qwen2.5-0.5B    | 13.2       | 13.6          | 6.4      |
> | Qwen2.5-1.5B    | 37.5       | 19.7          | 38.8      |
> | Qwen2.5-3B      | 22.9       | 39.4          | 49.1      |
>
>
> We observe that overall, a larger model follows the instruction better to follow the format and therefore, more data is kept.
>
>
> (ii) For token-level statistics, the SFT training split contains 1,280 trajectories, averaging about 2k tokens per trajectory, for a total of approximately 2.6 million tokens. During PPO training, we run 1,000 steps with ppo_mini_batch_size = 64. The total number of PPO tokens can be estimated as 1000 × 64 × 2k = 128 million tokens, which is roughly ≈50× the SFT scale.
>
>
> (iii) In terms of wall-clock cost, one complete SPA training run on a single environment takes roughly 30 minutes for SFT on a single GPU and 12 hours on 8 GPUs for RL. SFT introduces a negligible overhead for the training.

---

> ### Author Response · Authors · 2025-11-22
> **Response to Reviewer Piev (2)**
>
> **W4: The paper documents early exploration (Pass@k↑), then exploitation (Pass@k↓), but provides limited analysis on why SPA changes the diversity of trajectories**
>
>
> The paper documents early exploration (Pass@k↑), then exploitation (Pass@k↓) only for SPA training.
>
>
> When k is within a reasonable range, pass@k fundamentally evaluates constrained exploration capability: without an accurate transition kernel, stochastic rollouts often squander the budget k on hallucinations, deadlocks, or invalid actions. By learning the kernel, we constrain the policy to explore strictly within the true feasible manifold. This focused exploration enables more accurate planning during training. Thus, these two aspects represent different perspectives on the same underlying mechanism.
> We provide a case study here showing how the rollout becomes more accurate through world modelling.
>
> To illustrate how learning an accurate transition kernel expands the effective reachable set during model-based planning, we present a concrete example on Sokoban.
>
> Consider the following initial state (P is player, X is box and O is the target):
> ```
> ######
> ##___#
> ##_P_#
> #_OX##
> #____#
> ######
> ```
>
> Player (P) at (2,3); Box (X) at (3,3); Target (O) at (3,2)
>
> Without world modeling (Baseline): The model generates action sequence Down → Right → Down → Up → Left, predicting the following state:
> ```
> ######
> ##___#
> ##__X#
> #_OP##
> #____#
> ######
> ```
>
> However, this trajectory contains invalid actions (actions 2-3 violate movement constraints) and the predicted state is hallucinated—it violates the game's physics. The actual state after executing the valid subset of actions should be:
> ```
> ######
> ##___#
> ##___#
> #_OP##
> #__X_#
> ######
> ```
>
> More critically, subsequent rollouts from this hallucinated state continue to compound errors. For instance, the model may predict:
>
> Action: Down → Down → Right → Up
>
> Result:
> ```
> ######
> ##___#
> ##___#
> #_OP##
> #___X#
> ######
> ```
>
> The box is now trapped in a corner (an irreversible deadlock state), yet the model fails to recognize this. Subsequent rollouts waste the remaining budget on meaningless actions in this invalid state. When performing k stochastic rollouts, most samples diverge into similar hallucinated or deadlocked states, failing to explore genuinely diverse valid solutions.
> After SPA training: Starting from the same initial state, the model now produces valid, diverse trajectories:
>
> Rollout 1:
>
> Action: Left → Down → Down → Right → Up
>
> State:
> ```
> ######
> ##___#
> ##_X_#
> #_OP##
> #____#
> ######
> ```
>
> Player (P) at (3,3); Box (X) at (2,3); Target (O) at (3,2)
>
> Rollout 2:
>
> Action: Left → Up → Up → Right → Right
>
> State:
> ```
> ######
> ##__P#
> ##_X_#
> #_O_##
> #____#
> ######
> ```
>
> Player (P) at (1,4); Box (X) at (2,3); Target (O) at (3,2)
>
> Rollout 3:
>
> Action: Down → Left → Up → Left → Down
>
> Result: Goal achieved ✓
>
>
> The trained model maintains fidelity to true environment dynamics throughout all rollouts. All intermediate states remain within the valid manifold, enabling meaningful exploration of distinct solution paths. This demonstrates how an accurate transition kernel prevents cascading errors and allows k stochastic rollouts to genuinely cover k diverse valid strategies, rather than degenerating into the same hallucinated region.
> Quantitatively, SPA demonstrates substantially better exploration efficiency than the state estimation baseline. We measure effective exploration as the percentage of actions that move the box closer to the target (by Manhattan distance). While the baseline plateaus around 36%, SPA achieves up to 66.7% effective exploration rate at step 800, nearly doubling the baseline's performance. This gap highlights SPA's ability to learn goal-directed policies through world model-augmented self-play, rather than random exploration.
>
>
> | **Steps** | **0** | **200** | **400** | **600** | **800** | **1000** |
> |:--|:--:|:--:|:--:|:--:|:--:|:--:|
> | **State Estimation RL (%)** | 11.6 | 15.5 | 30.5 | 32.9 | 32.3 | 36.3 |
> | **SPA (Ours) (%)** | 8.2 | 20.4 | 49.9 | 45.9 | 66.7 | 50.5 |
>
>
> **Key Insight**: By constraining predictions to the true feasible manifold, SPA transforms the same budget k from wasteful hallucinations into effective exploration of genuinely reachable states. This directly explains why learning the transition kernel expands the effective reachable set and improves Pass@k—not by broadening exploration indiscriminately, but by ensuring every rollout contributes to discovering valid solutions.

---

> ### Author Response · Authors · 2025-11-22
> **Response to Reviewer Piev (3)**
>
> **Q1: How many SFT tokens are used relative to PPO tokens? What fraction of self-play samples are filtered for format issues?**
>
>
> For token-level statistics, The SFT training split contains 1,280 trajectories, averaging about 2k tokens per trajectory, for a total of approximately 2.6 million tokens. During PPO training, we run 1,000 steps with ppo_mini_batch_size = 64. The total number of PPO tokens can be estimated as 1000 × 64 × 2k = 128 million tokens, which is roughly ≈50× the SFT scale.
>
>
> ---
>
> **Q2: You mention fragility in stochastic settings; which component is most brittle (format compliance, prediction variance, critic learning)? Would modeling transition uncertainty (e.g., distributional next-state tokens or latent ensembles) help?**
>
>
> We think the fragility mainly stems from the difficulty of credit assignment in stochastic multi-turn settings. When multiple rollouts or dialogue turns are involved, the noise in outcomes makes it harder to trace which specific step, action, or generation contributed most to the final success or failure. As a result, the learning signal becomes diluted or misattributed, leading to instability in optimization.
>
>
> Modeling transition uncertainty (e.g., using distributional next-state representations or latent ensembles) could potentially mitigate this issue by explicitly capturing the variability in transitions, providing a smoother and more informative learning signal across turns.

---

### Official Review · Reviewer_56Et · 2025-11-01

**Soundness:** 2
**Presentation:** 3
**Contribution:** 2
**Rating:** 2
**Confidence:** 4

**Summary:**

This paper focuses on tackling the performance of language models under OOD scenarios by learning a world model via self play before policy finetuning. The authors highlight that by cold starting with SFT self play to learn the world model then use it to predict future states, their model can broaden state coverage and better align with real dynamics. They demonstrate that their method can increase both pass@1 and pass@k performance in multiple unfamiliar environments, showing the model's performance gain as well as sample diversity.

**Strengths:**

1. How to incorporate world model is a timely and important research direction to ground LLMs.

2. The paper is well motivated and clearly presented, tackling the OOD problem with world modeling is intuitive and reasonable.

3. Included detailed ablations demonstrating the effectiveness of learning transition modeling and better state representation.

**Weaknesses:**

1. The idea of self play explored in previous literature [1, 2] refers to a multi-agent setup, where LLM agents are used as different players in the game to interact with each other. This work, on the other hand, rolls out the base model in the environment for data collection and claims it to be self play. This is not self play widely considered in previous work in LLM or multi-agent RL[3, 4].

[1] Chen, Zixiang, et al. "Self-play fine-tuning converts weak language models to strong language models." arXiv preprint arXiv:2401.01335 (2024)

[2] Cheng, Pengyu, et al. "Self-playing adversarial language game enhances llm reasoning." Advances in Neural Information Processing Systems 37 (2024): 126515-126543

[3] Silver, David, et al. "Mastering chess and shogi by self-play with a general reinforcement learning algorithm." arXiv preprint arXiv:1712.01815 (2017)

[4] Heinrich, Johannes, and David Silver. "Deep reinforcement learning from self-play in imperfect-information games." arXiv preprint arXiv:1603.01121 (2016).

**Questions:**

1. Could the authors explain why learning the transition kernel expands reachable sets under k stochastic rollouts in Lines 90-91? Learning a transition kernel would help more with accurate planning instead of increasing coverage of reachable sets, which would also help lifting Pass@k. Also, this seems to overlap with the previous claim for broader state coverage.


2. In Section 4.2, the authors test the effect of using ground truth coordinates as an additional state representation by replacing them with iid random coordinates and show that training collapsed. Random coordinates are likely to mismatch the actual state observation, thus no surprise that the training would collapse. A more meaningful ablation would be to compare the performance with just raw state observations.


3. Related work should include world modeling and self-play for LLM.

---

> ### Author Response · Authors · 2025-11-22
> **Response to Reviewer 56Et (1)**
>
> Thanks for your comments and advice, we are pleased that you find our method is **well-motivated** and **intuitive and reasonable**. And our experiment is **detailed** and learning transition modeling is **effective**. We address your concerns below.
>
> **W1: This is not self play widely considered in previous work in LLM or multi-agent RL [3, 4].**
>
> Thank you for the reminder. We would like to clarify if our description and terminology have caused any confusion for the reviewer.
> Self-play is a general reinforcement learning paradigm where agents learn through autonomous interaction, not limited to multi-agent competition. For instance, MuZero explicitly uses the term “self-play data generation” to describe single-agent data collection through interaction with the environment, without access to ground-truth dynamics or rules [1] — “For each board game, we used 16 TPUs for training and 1000 TPUs for self-play” (Appendix G). Similarly, in [2], the authors introduce SelfPlayer, a data-efficient single-agent self-play exploration algorithm, where the agent learns from its own past experience.
> However, if this terminology causes any confusion, we are open to replacing it with alternative terms such as “self-experience” or “self-exploration” for clarity.
>
> **References**
>
> [1] Schrittwieser et al., Mastering Atari, Go, Chess and Shogi by Planning with a Learned Model, Nature, 2020.
>
> [2] Srinivas et al.,Data-Efficient Exploration with Self Play for Atari, URL, 2021
>
> ---
>
> **Q1: Could the authors explain why learning the transition kernel expands reachable sets under k stochastic rollouts in Lines 90-91?**
>
> We thank the reviewer for this insightful question. Improving prediction accuracy and expanding the effective reachable set are intrinsically linked. When k is within a reasonable range, pass@k fundamentally evaluates constrained exploration capability: without an accurate transition kernel, stochastic rollouts often squander the budget k on hallucinations, deadlocks, or invalid actions. By learning the kernel, we constrain the policy to explore strictly within the true feasible manifold. This focused exploration enables more accurate planning during training. Thus, these two aspects represent different perspectives on the same underlying mechanism.
>
>
> To illustrate this mechanism, we present a concrete example on Sokoban planning. Consider the following initial state (P is player, X is box and O is the target):
>
> ```
> ######
> ##___#
> ##_P_#
> #_OX##
> #____#
> ######
> ```
>
> Player (P) at (2,3); Box (X) at (3,3); Target (O) at (3,2)
>
> Without world modeling (Baseline): The model generates action sequence Down → Right → Down → Up → Left, predicting the following state:
>
> ```
> ######
> ##___#
> ##__X#
> #_OP##
> #____#
> ######
> ```
>
> However, this trajectory contains invalid actions (actions 2-3 violate movement constraints) and the predicted state is hallucinated—it violates the game's physics. The actual state after executing the valid subset of actions should be:
>
> ```
> ######
> ##___#
> ##___#
> #_OP##
> #__X_#
> ######
> ```
>
> More critically, subsequent rollouts from this hallucinated state continue to compound errors. For instance, the model may predict:
>
> Action: Down → Down → Right → Up
>
> Result:
> ```
> ######
> ##___#
> ##___#
> #_OP##
> #___X#
> ######
> ```
>
> The box is now trapped in a corner (an irreversible deadlock state), yet the model fails to recognize this. Subsequent rollouts waste the remaining budget on meaningless actions in this invalid state. When performing k stochastic rollouts, most samples diverge into similar hallucinated or deadlocked states, failing to explore genuinely diverse valid solutions.
> After SPA training: Starting from the same initial state, the model now produces valid, diverse trajectories:
>
> Rollout 1:
>
> Action: Left → Down → Down → Right → Up
> ```
> State:
> ######
> ##___#
> ##_X_#
> #_OP##
> #____#
> ######
> ```
>
> Player (P) at (3,3); Box (X) at (2,3); Target (O) at (3,2)
>
>
> Rollout 2:
>
> Action: Left → Up → Up → Right → Right
>
> ```
> State:
> ######
> ##__P#
> ##_X_#
> #_O_##
> #____#
> ######
> ```
>
> Player (P) at (1,4); Box (X) at (2,3); Target (O) at (3,2)
>
> Rollout 3:
>
> Action: Down → Left → Up → Left → Down
>
> Result: Goal achieved ✓
>
> The trained model maintains fidelity to true environment dynamics throughout all rollouts. All intermediate states remain within the valid manifold, enabling meaningful exploration of distinct solution paths. This demonstrates how an accurate transition kernel prevents cascading errors and allows k stochastic rollouts to genuinely cover k diverse valid strategies, rather than degenerating into the same hallucinated region.

---

> ### Author Response · Authors · 2025-11-22
> **Response to Reviewer 56Et (2)**
>
> (Continued for Q1)
>
> Quantitatively, SPA demonstrates substantially better exploration efficiency than the state estimation baseline. We measure effective exploration as the percentage of actions that move the box closer to the target (by Manhattan distance). While the baseline plateaus around 36%, SPA achieves up to 66.7% effective exploration rate at step 800, nearly doubling the baseline's performance. This gap highlights SPA's ability to learn goal-directed policies through world model-augmented self-play, rather than random exploration.
>
>
> | **Steps** | **0** | **200** | **400** | **600** | **800** | **1000** |
> |:--|:--:|:--:|:--:|:--:|:--:|:--:|
> | **State Estimation RL (%)** | 11.6 | 15.5 | 30.5 | 32.9 | 32.3 | 36.3 |
> | **SPA (Ours) (%)** | 8.2 | 20.4 | 49.9 | 45.9 | 66.7 | 50.5 |
>
>
> **Key Insight**: By constraining predictions to the true feasible manifold, SPA transforms the same budget k from wasteful hallucinations into effective exploration of genuinely reachable states. This directly explains why learning the transition kernel expands the effective reachable set and improves Pass@k—not by broadening exploration indiscriminately, but by ensuring every rollout contributes to discovering valid solutions.
>
> ---
>
> **Q2: A more meaningful ablation would compare performance with just raw state observations.**
>
> Thank you for the suggestion. We have already included this experiment under our State Estimation method (the yellow line in Figure 2 and all State Estimation RL rows in Table 2). In addition, our ablation study—where we replaced the coordinates with random values—is not entirely a “no-surprise” case. Random coordinates could, in principle, act as a reasoning format that bridges symbolic inputs with reasoning chains, potentially triggering the model’s tendency to ground on coordinate-like cues. However, the random coordinates fail to help, indicating that the model cannot perform spatial reasoning even when such trigger patterns are present, and it needs to ground on accurate coordinates to reason correctly.
>
> ---
> **Q3: Related work should include world modeling and self-play for LLM.**
>
> Thanks for the suggestion. We will expand Section 6 with coverage of World Modeling and Self-Play for LLMs:
>
> **World Modeling and Self-Play for LLMs**: World models learn environment dynamics to enable model-based planning ([1,2,3]), with recent work extending to LLM agents ([4]). While traditional self-play involves agents competing against themselves in multi-agent settings ([5]), recent LLM work has adapted this paradigm: SPIN ([6]) distinguishes self-generations from reference data, Self-Rewarding ([7]) generates self-rewards, and SPAG ([8]) uses adversarial self-play for reasoning. Our approach differs by using "self-play" to denote autonomous trajectory generation through environment rollouts—a usage established by MuZero ([9]) for single-agent Atari environments and SelfPlayer ([10]) for single-agent exploration.
>
> We’ve also expanded our paper in lines 477-485.
>
>
> **References**
>
> [1] Ha & Schmidhuber, "World Models," NeurIPS 2018
>
> [2] Hafner et al., "Mastering Atari with Discrete World Models," ICLR 2021
>
> [3] Micheli et al., "Transformers are Sample-Efficient World Models," ICLR 2023
>
> [4] Hao et al., "Learning to Model the World with Language," arXiv 2024
>
> [5] Silver et al., "Mastering the game of Go without human knowledge," Nature 2017
>
> [6] Chen et al., "Self-Play Fine-Tuning Converts Weak Language Models to Strong Language Models," arXiv 2024
>
> [7] Yuan et al., "Self-Rewarding Language Models," arXiv 2024
>
> [8] Pang et al., "Self-playing Adversarial Language Game Enhances LLM Reasoning," NeurIPS 2024
>
> [9] Schrittwieser et al., "Mastering Atari, Go, chess and shogi by planning with a learned model," Nature 2020
>
> [10] Nikishin et al., "Data-Efficient Exploration with Self Play for Atari," ICLR Workshop 2021

---

### Meta-Review · Area_Chair_eYhh · 2026-01-06

**Summary:**

This work aims to address out-of-distribution (OOD) challenges that arise when fine-tuning LLM-based agents in shifted environments. The authors claim that explicitly training the model to learn state transitions improves the agent’s exploration capability. The main concerns raised by reviewers relate to the limited experimental settings, as most evaluations are conducted on text-based games. In addition, the reliance on handcrafted state representations limits the generality of the proposed method.

**Reviewer Concerns:**

**Unaddressed concerns:**

**Reviewer 56Et:** use of the term “self-play”.

**Reviewer Piev:** benchmarks limited to text-based games and lack of in-depth analysis.

**Reviewer jZgi:** experiments are overly simplistic.

**Reviewer FybG:** experiments restricted to synthetic tasks and reliance on hand-crafted state representations.

**Reviewer Scores:**

**Reviewer 56Et:** 2 → 4

**Reviewer Piev:** 6 → 6

**Reviewer jZgi:** 6 → 6

**Reviewer FybG:** 4 → 4

---

### Decision · Program_Chairs · 2026-01-26

Reject